# Development of an RNA Nanostructure for Effective *Botrytis cinerea* Control through Spray-Induced Gene Silencing without an Extra Nanocarrier

**DOI:** 10.3390/jof10070483

**Published:** 2024-07-14

**Authors:** Fangli Wu, Ling Yan, Xiayang Zhao, Chongrun Lv, Weibo Jin

**Affiliations:** 1Key Laboratory of Plant Secondary Metabolism and Regulation of Zhejiang Province, College of Life Sciences and Medicine, Zhejiang Sci-Tech University, Hangzhou 310018, China; wfl@zstu.edu.cn (F.W.); 202130802168@mails.zstu.edu.cn (L.Y.); zstuzxy602@126.com (X.Z.); 202130802142@mails.zstu.edu.cn (C.L.); 2Zhejiang Sci-Tech University Shaoxing Academy of Biomedicine, Shaoxing 312366, China

**Keywords:** RNA nanoparticle, fungicide, nanocarrier free, SIGS, *Botrytis cinerea*

## Abstract

Spray-induced gene silencing represents an eco-friendly approach for crop protection through the use of double-stranded RNA (dsRNA) to activate the RNA interference (RNAi) pathway, thereby silencing crucial genes in pathogens. The major challenges associated with dsRNA are its limited stability and poor cellular uptake, necessitating repeated applications for effective crop protection. In this study, RNA nanoparticles (NPs) were proposed as effectors in plants and pathogens by inducing the RNAi pathway and silencing gene expression. RNA structural motifs, such as hairpin-loop, kissing-loop, and tetra-U motifs, were used to link multiple siRNAs into a long, single-stranded RNA (lssRNA). The lssRNA, synthesized in *Escherichia coli*, self-assembled into stable RNA nanostructures via local base pairing. Comparative analyses between dsRNA and RNA NPs revealed that the latter displayed superior efficacy in inhibiting spore germination and mycelial growth of *Botrytis cinerea*. Moreover, RNA NPs had a more robust protective effect on plants against *B. cinerea* than did dsRNA. In addition, RNA squares are processed into expected siRNA in plants, thereby inhibiting the expression of the target gene. These findings suggest the potential of RNA NPs for use in plant disease control by providing a more efficient and specific alternative to dsRNA without requiring nanocarriers.

## 1. Introduction

Exogenous double-stranded RNAs (dsRNAs) or small interfering RNAs (siRNAs) can potentially be internalized by plants and subsequently transported into pathogens, including fungi, oomycetes, nematodes, and insects. Alternatively, they can directly enter pathogenic organisms, thereby accomplishing disease prevention and control through RNA interference (RNAi), targeting vital genes of pathogenic microorganisms [1]. This phenomenon, known as spray-induced gene silencing (SIGS), is recognized as another effective disease control technique based on the RNAi pathway after host-induced gene silencing. SIGS can prevent diseases by inhibiting specific pathogen genes without interfering with host gene expression. RNAi-based biopesticides have gained considerable attention in recent years due to their versatile applicability against various pathogens or pests and their environmentally friendly attributes [2]. However, certain shortcomings of dsRNA, such as its low RNAi induction efficiency and limited protection duration, pose urgent challenges when directly applying naked dsRNA or siRNA. This is primarily due to the inherent instability, rapid degradation, and poor uptake of RNA by plants [3,4]. Therefore, the utilization of nanoparticles (NPs) as carriers for dsRNA/siRNAs is highly promising for SIGS applications [5]. Notable examples include layered double hydroxide clay nanosheets [6], guanidine-containing polymers [7], and liposome complexes [8]. Nevertheless, recent studies have highlighted the potential risks associated with NPs, including unintended harm to nontarget organisms, accumulation through transportation and bioaccumulation, and interactions with other environmental contaminants and dissolved organic matter, potentially resulting in greater environmental damage [9,10,11].

RNA NPs, characterized by specific three-dimensional structures and robust stability [12,13,14], can be designed and assembled through bottom-up self-assembly using structural motifs. Typically, RNA NPs serve as structural scaffolds for delivering biologically active molecules. Recent evidence has demonstrated that the structural motifs within RNA NPs can also serve as substrates for Dicer, triggering the RNAi mechanism to silence target gene expression [15,16,17,18,19,20]. Consequently, we propose the application of siRNAs targeting key genes of pathogenic microorganisms as structural motifs to construct RNA nanostructures, enhancing both the stability and efficiency of siRNA delivery.

In this study, we selected a square RNA nanostructure, as reported by Li et al. [21], to design an antifungal RNA fungicide. This was achieved by linking multiple siRNAs targeting four virulence genes (*Bcin12g06230.1* (*Dicer-Like 1*, *DCL1*), *Bcin13p01840.1* (*Peptidyl-prolyl cis-trans isomerase 10*, *PPI10*), *Bcin04p04920.1* (*N-myristoyltransferase 1*, *NMT1*) and *Bcin15p02590.2* (*putative adenylate cyclase*, *BAC*)) which was validated as an effective target gene for the RNAi-based management of *Botrytis cinerea* [1,22]. Subsequently, the anti-*B. cinerea* RNA squares were synthesized in *E. coli* using IPTG induction. Finally, we evaluated the anti-*B. cinerea* activity of the RNA square by assessing its inhibitory effect on spore germination, mycelial growth, and virulence of *B. cinerea*. This study presents a novel strategy to increase the stability and specificity of dsRNA fungicides, as well as novel directions for pathogen control via nanocarrier-free SIGS.

## 2. Materials and Methods

### 2.1. Preparation of Plant Materials and Inoculation of Botrytis cinerea

Tobacco plants (*Nicotiana benthamiana*) were conventionally cultivated in a greenhouse under conditions featuring a 16 h light and 8 h dark photoperiod, with temperatures maintained at approximately 24 °C. Potato dextrose agar (PDA) served as the medium for the cultivation of *B. cinerea* strain B05.10. Conidiospores were harvested from two-week-old PDA cultures, suspended in milli-Q water, and adjusted to a concentration of 5 × 10^6^ conidiospores/mL for subsequent bioassays.

### 2.2. Design of a B. cinerea-Targeted RNA Nanoparticle

The online program called Designer of Small Interfering RNA (DSIR) was utilized to design 21 nt siRNAs, ensuring a score higher than 90 and fewer than four consecutive identical nucleotides [23]. To assess potential off-target effects, the siRNAs were subjected to a NCBI BLASTn search against the tomato genome utilizing default parameters, except for a word size of 7 [24]. After eliminating off-target siRNAs, seven siRNAs for the target gene were randomly selected and subsequently concatenated using tetra-U helix linking motifs [21], hairpin loops (5′-UCCG-3′), and kissing loops (5′-AAGGAGGCA-3′, 5′-AAGCCTCCA-3′), resulting in the formation of a square RNA nanostructure. The secondary structure of the RNA square was confirmed using the RNAfold program [25].

### 2.3. Construction of RNA Squares and dsRNA Expression Vectors

Initially, the 5′-end of the DNA sequence encoding the RNA square was appended with the T7 promoter (5′-TCTAATACGACTCACTATA-3′). Subsequently, two restriction enzymes, namely, Bgl II and Bpu 1102, were introduced at both ends of this sequence (Appendix A). The complete sequence was chemically synthesized by Zhejiang RNA-Direct Nanotech Co., Ltd. (Shaoxing, China) and subsequently inserted into the pET28a plasmid to facilitate the synthesis of the RNA square within *E. coli*.

For the synthesis of a dsRNA targeting *DCL1*, *PPI10*, *Nmt1*, and *BAC*, a 100 bp gene-specific sequence was obtained from each of the four target genes (*DCL1*, *PPI10*, *NMT1*, and *BAC*) based on blast searching against the mRNA library of *B. cinerea*. The specific sequences of each target gene are as follows:

*PPI10*: 5′-GGA GAA GGT GCT GAT TGA GGG GGT GAC GGT GCA TGC TAA TCC GCT TGC GGG CTG AGC TTT AAT GTG TGG CTT TGG GCT GGG ATT TAG GTT TGA GAT GGT G-3′.

*NMT1*: 5′-CCC TCG TTC TGA TGT TCA AAT GCT CCG TAT TGA TGA GCT TGC TGA GCT TGG ATG TTG TTG CTT CTG TTC CAT TTT GTA TAT TGC AAA TGA TGC ATT CAT T-3′.

*BAC*: 5′-ATC AAT GAA GAG AGG AGC TCC AAT GCA GGT GAA GAC GGA TAG ATC AAG TAG GGT AGA AAT TGA TGC TCA AGT TTC GCC GAC GTC TGC CGG ACC TCG AAA T-3′.

*DCL1*: 5′-CTT CTC GAG ATA AGA TAC CTT CTG CAT CTG GCA ACG GAG ATG CTA TAG CAG ATG TTA GCA GTG GTT ACC TCA AAC AGG CTA CCG TAT CTT CTC ATT CTG C-3′.

These four sequences are all 100 nt in length and then were connected into a 400 bp sequence (Appendix A). Subsequently, two restriction enzyme sites, namely, Xba I and Hind III, were introduced at both ends of this sequence. The complete sequence was chemically synthesized and inserted into L4440 plasmid using Xba I and Hind III restriction enzymes. The positive recombinant plasmid is capable of generating long dsRNA molecules within *E. coli*.

### 2.4. In Vivo Synthesis of the RNA Square and dsRNA

The recombinant plasmid was subsequently introduced into *E. coli* HT115 (DE3), after which RNA transcription was induced by 1 mmol/L IPTG once the OD600 reached 0.4. After a 4 h incubation period, 1 mL of cells was harvested and reconstituted in 100 µL of Rnase-free solution (10 mmol/L Tris-HCl, pH 8.0, 10 mmol/L MgSO_4_); subsequently, a mixture of phenol:chloroform:isoamyl alcohol (25:24:1) was added to an equal volume to lyse the cells. The aqueous component of the total cell lysates was purified with 1/2 volume of chloroform to eliminate any residual phenol. The resulting aqueous solution containing RNA square or dsRNA was then directly subjected to examination via agarose gel electrophoresis. RNA concentration was calculated by measuring the absorbance of 1 µL of the aqueous solution at 260/280 nm using the NanoDrop 2000 (Thermo Fisher Scientific, Waltham, MA, USA). RNA extracted from the induced *E. coli* harboring the empty plasmid that served as NC-RNA.

### 2.5. Conidiospore Germination Assays

Conidiospore germination was assessed using the cellophane strip method, as previously described by Bilir et al. [26]. In brief, cellophane strips measuring 1.5 cm^2^ were sterilized under high pressure and then positioned on Murashige and Skoog (MS) media (4.3 g/L MS basal salts supplemented with sucrose 1% (*m*/*v*), adjusted to pH 5.7, and solidified with 1.5% agar) within Petri dishes. Subsequently, 80 µL of a 100-fold diluted spore solution (5 × 10^6^ spores/mL) was pipetted onto the cellophane strips. Next, 2 µL of RNA was added to the spore solution and thoroughly mixed to achieve a final RNA concentration of 100 ng/µL. After incubating for a 12 h:12 h photoperiod at 24 °C for 24 h, the spores were examined using a light microscope.

### 2.6. Inhibitory Effects of the RNA Square on B. cinerea Mycelial Growth

RNA square or dsRNA was added to PDA media at a final concentration of 100 ng/μL before the media solidified and then poured into 9 cm Petri dishes containing 25 mL PDA. The PDA plates without RNA (no RNA) and with NC-RNA were used as controls. Each of the RNAs were set up with four parallel repetitions. Subsequently, 4 mm diameter agar plugs containing active mycelium of *B. cinerea* from a 7-day-old PDA culture were inoculated at the center of the RNA-treated plates and cultured at 24 °C. When the colonies of the NC-RNA group completely covered the plate, the colony diameter of each plate was measured, and the inhibition rate was calculated by the following formula: IR (%) = 100% × (Dc − Dt)/Dc, where IR represents the inhibition rate, Dc represents the average diameter of the NC-RNA group, and Dt represents the average diameter of the dsRNA- or RNA square-treated group [27].

### 2.7. Bioassay of the RNA Square against B. cinerea

Tobacco plants with uniform growth were selected for bioassay, and each of them was manually sprayed with 2 mL of RNA solution at a concentration of 100 ng/µL. These RNA-sprayed plants were then grown in the greenhouse under a 16:8 h photoperiod at 24 °C. Each of the RNAs were set up with five parallel repetitions. At seven days postspraying (dps), a leaf per plant with uniform size was picked from each RNA-sprayed plant for *B. cinerea* inoculation. Subsequently, 4 mm diameter agar plugs containing active mycelium of *B. cinerea* from a 7-day-old PDA culture were inoculated on each leaf surface. The leaves without RNA treatment (no RNA) were inoculated with a mycelia agar plug as a negative control. Incubation was at 24 °C under a 16:8 h photoperiod and 85% humidity. Subsequently, lesion sizes on the pathogen-infected plant materials were measured at two days postinoculation (dpi).

### 2.8. Total RNA Extraction and Quantitative RT–PCR (RT-qPCR)

To detect the expression effects of the target gene in *B. cinerea*, inoculation with *B. cinerea* spores was conducted on 4-week-old tobacco plants. Five leaves per plant were pricked, and each leaf was inoculated with 10 μL of the *B. cinerea* spore solution (5 × 10^6^ spores/mL). Each inoculum was sprayed with 2 mL of RNA square or dsRNA (100 ng/µL) at 12 h after inoculation. These plants were cultured at 24 °C with a 16:8 h photoperiod and 85% humidity. Leaves from 5 individual RNA-sprayed plants were harvested at 1, 4, 7, 10, and 15 days postspraying (dps) for RNA extraction. The same experiment was performed without RNA treatment as a control. Total RNA extraction was performed following the protocol outlined by Meng et al. [28]. In brief, the total RNA was isolated using TRIzol, treated with RNase-free Dnase, and quantified using a NanoDrop ND-1000 Spectrophotometer. For poly(A) RNA, equal amounts of total RNA (1 µg) were reverse-transcribed at 42 °C utilizing SuperScript III Reverse Transcriptase (Invitrogen, Waltham, MA, USA) and 2.5 µM Oligo(dT_18_). A parallel reaction lacking reverse transcriptase was conducted as a control to confirm the absence of genomic DNA in subsequent steps.

SYBR Green PCR was carried out using a qTOWER3G fluorescence quantitative PCR instrument (Jena, Germany). Briefly, 2 µL of cDNA template was combined with 12.5 µL of 2× SYBR Green PCR master mix (Takara, Dalian, China), 1 µM specific primers, and ddH_2_O to reach a final volume of 25 µL. The reactions involved amplification for 10 s at 95 °C, followed by 40 cycles of 95 °C for 10 s and 60 °C for 30 s. All the reactions were conducted in triplicate, and controls (no template and no RT) were included for each gene. The threshold cycle (Ct) values were automatically determined using a qTOWER3G fluorescence quantitative PCR instrument (Jena, Germany). Fold changes were calculated employing the 2^−ΔΔCt^ method, where ΔΔCt= (Ct_target_ − Ct_inner_)_Infection_ − (Ct_target_ − Ct_inner_)_control_ [29]. The oligos used in this study are detailed in Appendix A.

### 2.9. Stem-Loop RT-qPCR

Stem-loop RT–qPCR was employed to verify siRNAs [30]. A pulsed reverse transcription reaction (16 °C for 30 min, followed by 60 cycles at 30 °C 30 s, 42 °C 30 s, and 50 °C for 1 min and for 5 min at 85 °C) [30] was performed using Takara PrimeScript RT reagent kit (RR037A, Takara, Dalian, China) with siRNA-specific stem-loop RT primers. Stem-loop qPCR reaction (95 °C for 10 s, followed by 60 cycles at 95 °C for 10 s, 60 °C for 10 s, and 72 °C for 10 s) was performed using 2× SYBR Green PCR master mix (Takara), with specific primers and qTOWER3G fluorescence quantitative PCR instrument (Jena, Germany).

### 2.10. Statistical Analysis

Statistical analysis was performed using GraphPad Prism software version 8.0. The normality distribution was assessed using Shapiro–Wilk test. One-way ANOVA followed by Tukey’s test was used to determine whether there was a statistically significant difference in different independent groups.

## 3. Results

### 3.1. Design of a B. cinerea-Targeted Square RNA Nanostructure

To construct a *B. cinerea*-targeted RNA nanostructure, we selected four virulence genes, *DCL1*, *PPI10*, *Nmt1*, and *BAC,* which have been verified to be effective target genes for RNAi-based management of *B. cinerea* [1,22]. To design *B. cinerea*-targeted siRNAs, we utilized DSIR and randomly screened out seven siRNAs for these four target genes (Table 1 and Appendix A). These seven siRNAs served as structural motifs for the assembly of RNA NPs via a square-like configuration utilizing hairpin loops, kissing loops, and the tetra-U motif (Figure 1a). This assembly was validated using the RNAfold program and named RNA square (Figure 1b and Appendix A). The coding DNA sequence of the RNA square is provided in Appendix A.

### 3.2. Production of RNA Nanoparticles

To generate the RNA square within *E. coli*, DNA templates encoding the RNA square were synthesized and subsequently inserted into the pET28a plasmid, which were subsequently introduced into *E. coli* HT115. Additionally, we constructed a plasmid for the synthesis of *B. cinerea*-targeted dsRNA in *E. coli*. The expression and accumulation of both the RNA square and dsRNA within *E. coli* were assessed using native agarose gel electrophoresis. The results revealed the presence of specific target bands in the lanes loaded with dsRNA and RNA squares, indicating successful synthesis and accumulation of the target RNAs in HT115 cells (Figure 2).

### 3.3. The Anti-B. cinerea Activity of the RNA Square

To evaluate the anti-*B. cinerea* activity of the RNA square concerning the growth and development of *B. cinerea*, we initiated our assessment by examining its effect on spore germination. The results indicated that the NC-RNA had no discernible effect on spore germination compared to spore germination without RNA treatment (no RNA). dsRNA exhibited only partial inhibition of germination. Remarkably, the RNA square completely inhibited *B. cinerea* spore germination (Figure 3a). Furthermore, we scrutinized the inhibitory influence of the RNA square on the mycelial growth of *B. cinerea*. The results illustrated that the average colony diameter of the *B. cinerea* on the PDA plate without RNA (no RNA) and with NC-RNA were approximately 85.7 mm and 84.4 mm, respectively, whereas the colony size decreased to approximately 44.8 mm in the presence of dsRNA, reflecting an inhibition rate of approximately 48%. Notably, in the RNA square-treated plate, the diameter of the *B. cinerea* colonies was only 19.5 mm, indicating a remarkable inhibition rate of approximately 77% (Figure 3b,c). These findings strongly suggest that, in comparison to dsRNA, the RNA square exhibits superior efficacy in inhibiting both spore germination and mycelial growth of *B. cinerea*.

### 3.4. Bioassay of the RNA Square in Tobacco Leaves against B. cinerea

To assess the antifungal efficacy of the RNA square against *B. cinerea*, a 2 mL solution of RNA square at a concentration of 100 ng/µL was applied by spraying onto whole tobacco plants. Subsequently, *B. cinerea* were inoculated onto the leaves at 7 dps. After two days of inoculation, the findings revealed that the average diameters of the necrotic regions were approximately 17.0 mm and ~15.5 mm for those treated without RNA (no RNA) and with NC-RNA, respectively. In contrast, the diameters were significantly smaller, approximately 13.5 and ~7 mm, in the leaves treated with the dsRNA and RNA square, respectively (Figure 4a,b). These results strongly indicate that the RNA square exhibits superior antifungal activity compared to dsRNA.

### 3.5. siRNA Generation and RNAi Efficiency of the RNA Squares

To compare the inhibitory efficiency of the RNA square and dsRNA on the target gene, both dsRNA and RNA squares were initially applied by spraying onto *B. cinerea*-inoculated tobacco leaves. Subsequently, quantitative RT–PCR was employed to examine the expression profiles of the *BcDCL1* at various time points following RNA application. The findings demonstrated that, on plant leaves treated with the RNA square, the expression of *BcDCL1* exhibited an inhibition of approximately 50% within the initial 10 days, with particularly robust inhibition of approximately 75% at 4 dps. In contrast, the inhibitory effect of *BcDCL1* on dsRNA-treated leaves were lower than that on RNA square-treated leaves. Moreover, the inhibitory effect of *BcDCL1* was less than 30% or even disappeared at 7 dps and later times. At 7 dps and subsequent time points, the expression levels of *BcDCL1* on dsRNA-treated leaves reached more than 70% or even higher compared to those on control leaves (Figure 5a). In addition, the expression level of *BcDCL1-siR1*, a *BcDCL1*-targeting siRNA used as a structural motif in RNA square, was examined using stem-loop RT-qPCR. Results showed that the expression level of *BcDCL1-siR1* in RNA square-treated *B. cinerea* was highest at 4 dps and then gradually decreased at subsequent time points (Figure 5b). These results indicated that the RNA square taken up by *B. cinerea* may be completely processed into siRNA within 4 days, resulting in the highest siRNA level at 4 dps. Correspondingly, the level of *BcDCL1* was the lowest at 4 dps in *B. cinerea* with RNA square treatment. Our results confirmed that the RNAi efficiency induced by the RNA square is notably greater than that induced by dsRNA.

## 4. Discussion

Contrary to the conventional belief that RNA is inherently unstable and susceptible to degradation, RNA NPs with specialized structures exhibit remarkable thermodynamic and biological stability, surpassing even DNA NPs [31]. The discovery of thermodynamically stable RNA NPs offers compelling evidence for the potential therapeutic applications of RNA NPs in vivo. Studies by Nakashima et al. demonstrated that RNAi induced by small RNA (sRNA) loaded onto three-way junction RNA NPs can persist for up to five days [32]. Additionally, Afonin et al. reported that RNAi induced by siRNA loaded onto RNA NPs can persist for nine days, with an RNAi efficiency sixfold greater than that of free siRNA [33]. Moreover, recent research has revealed that RNA NPs themselves can trigger RNAi. Jang et al. reported that three-way RNA NPs can induce RNAi through self-disintegration [34]. Jedrzejczyk et al. demonstrated that triangular RNA nanostructures not only serve as substrates for Dicer enzymes to initiate gene silencing but also prolong the regulation of gene expression [19]. In this study, we further confirmed that the use of siRNAs as structural motifs for RNA NPs not only suppressed the growth and virulence of *B. cinerea* through the RNAi pathway but also achieved greater efficiency and longer-lasting effects than did the use of dsRNA. SIGS is a novel plant protection strategy by inducing the RNAi pathway in fungal pathogens and has been successfully applied through a variety of methods such as spraying, infiltration, root soaking, leaf propagation, and injection [35,36]. Exogenous dsRNA can be directly taken up by fungal pathogens, and/or taken up by the host plants first, transferred into fungal cells, and can subsequently induce the fungal RNAi machinery [37]. Although promising, sRNA and dsRNA molecules have a short period of environmental stability and variable uptake rates in pathogens [1]. To improve the efficacy of SIGS, recent advances have been made in nanoparticle-based protection of RNAs that drastically increase the feasibility of RNA-based crop protection strategies [38]. For example, compared with naked dsRNA, dsRNA loaded into BioClay NPs increased the protection window against *B. cinerea* from 1 to 3 weeks on tomato leaves and from 5 to 10 days on fruit [39]. dsRNA encapsulated with lipid NPs can also extend the protection window to 10 days against *B. cinerea* on tomato and grape fruits [40]. In this study, we developed an RNA square that uses siRNAs as the structural motifs of RNA NPs to improve the stability and uptake efficiency of siRNA. Compared with dsRNA, RNA square could provide stronger and longer-lasting protection for tobacco against *B. cinerea*. The expression pattern of the target gene suggests that RNA square could have the ability to produce siRNA and induce RNAi more persistently than dsRNA.

Generating sufficient quantities of dsRNA is another barrier to the application of SIGS in the field. Traditional dsRNA production methods in the laboratory are costly and produce only limited amounts of dsRNA, making them impractical for large-scale application needs [41]. To address the cost challenges associated with RNA nanoparticle synthesis, Li et al. [21] proposed a mass production strategy for RNA in *E. coli.* This strategy involves designing a single-stranded RNA molecule capable of self-assembly and folding into RNA NPs. Like in the case of foreign protein expression, these designed RNA nanostructures are synthesized within *E. coli.* The results demonstrated that double-square structured RNA NPs could be effectively accumulated in *E. coli* BL21 (DE3), accounting for 11.2% of the total RNA [14]. In our study, we utilized another *E. coli* HT115 strain, which is a mutant of *Rnase III*, to synthesize RNA NPs in vivo. Our findings corroborate the robust stability of RNA NPs and their high-efficiency accumulation within *E. coli.*

## 5. Conclusions

SIGS represents an environmentally sustainable approach to crop protection. Nevertheless, the direct application of dsRNA/siRNA often yields low RNAi-inducing efficiency and provides only a brief protection period. In our investigation, we incorporated *B. cinerea*-targeting siRNAs into RNA NPs as structural motifs, thereby enhancing their stability and delivery efficiency. These siRNA-functionalized square RNA NPs have demonstrated the ability to inhibit target genes more efficiently and for a more prolonged duration than dsRNA. By optimizing the nanostructure, the stability of RNA NPs can be further improved to achieve a longer protection window, avoiding the potential environmental risks caused by the application of additional nanomaterials.

## Figures and Tables

**Figure 1 jof-10-00483-f001:**
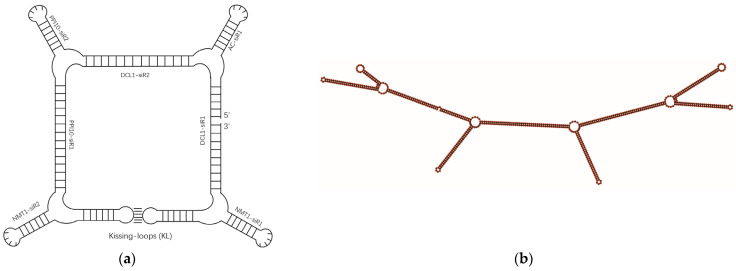
Design model of a square RNA nanoparticle (**a**) and its secondary structure confirmed by RNAfold (**b**).

**Figure 2 jof-10-00483-f002:**
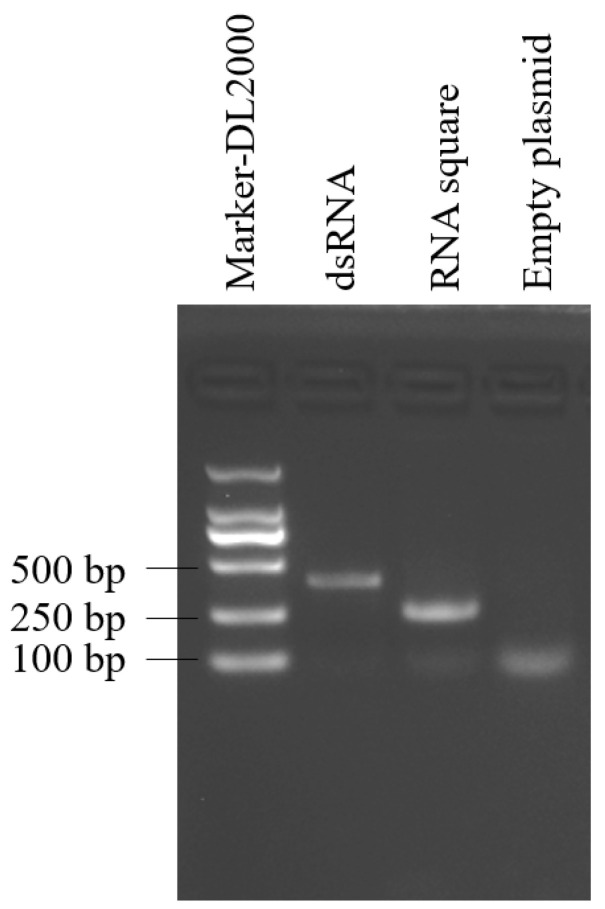
RNA extracted from induced *E. coli* harboring the empty plasmid or from the two recombinant plasmids.

**Figure 3 jof-10-00483-f003:**
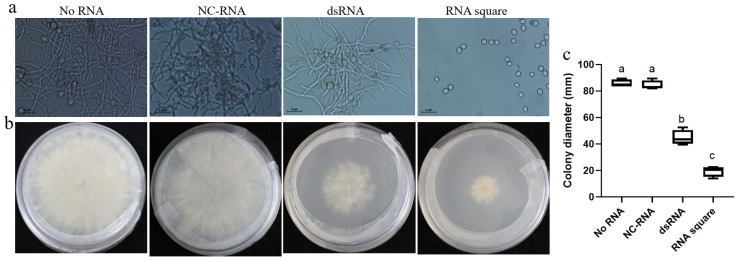
Antifungal activities of the square RNA nanostructures. (**a**) Germination effect of *B. cinerea* conidiospores treated with dsRNA and RNA squares. (**b**) Colony size of *B. cinerea* on a PDA plate containing dsRNAs or RNA squares. (**c**) The relative colony size of *B. cinerea*. Results are expressed as the means ± SEM of four biological replicates. The different letters indicate a significant difference as established by a one-way analysis of variance (ANOVA) with Tukey’s multiple comparisons test (*p* < 0.01).

**Figure 4 jof-10-00483-f004:**
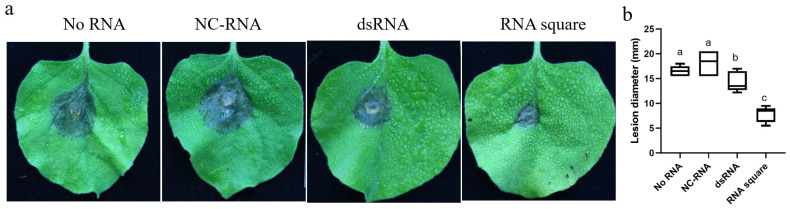
Protective efficacy of the RNA square against *B. cinerea* infection in tobacco plants. (**a**) Necrosis of *B. cinerea* on tobacco leaves. (**b**) The diameters of necrotic spots on tobacco leaves at 7 dps. The results are expressed as the means ± SEMs of five leaves. The different letters indicate a significant difference as established by a one-way analysis of variance (ANOVA) with Tukey’s multiple comparisons test (*p* < 0.01).

**Figure 5 jof-10-00483-f005:**
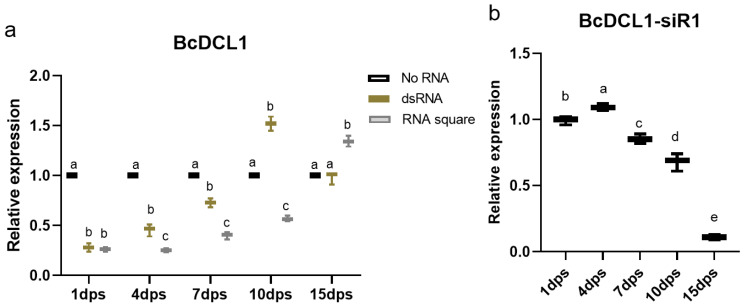
Expression levels of *BcDCL1* and *BcDCL1-siR1*. (**a**) Expression levels of *BcDCL1* in dsRNA or RNA square-sprayed tobacco leaves pretreated with *B. cinerea*. The *Actin* gene of *B. cinerea* was used as the internal control. The results are expressed as the means ± SEM of three biological replicates. Statistically significant differences according to Tukey’s test at the same treatment time are shown as different letters (*p* < 0.01). (**b**) Expression levels of *BcDCL1-siR1* in RNA square-sprayed tobacco leaves pretreated with *B. cinerea*. The *Actin* gene of *B. cinerea* was used as the internal control. The results are expressed as the means ± SEM of three biological replicates. The different letters indicate a significant difference as established by a one-way analysis of variance (ANOVA) with Tukey’s multiple comparisons test (*p* < 0.01).

**Table 1 jof-10-00483-t001:** Seven siRNAs used for the design of RNA square.

siRNA No.	Target Site	21nt Sense Sequence	21nt Antisense Sequence	Score
BcDCL1-siR1	2514–2534	GGUAGAUGCUAGAGAUAAUGU	AUUAUCUCUAGCAUCUACCGG	105.9
BcDCL1-siR2	5522–2542	CGGCAUACUUGUUCAUCUAUG	UAGAUGAACAAGUAUGCCGGA	101.9
BcPPI10-siR1	491–519	GCAUCUCGAUGGUCAGAAUAC	AUUCUGACCAUCGAGAUGCGG	96.1
BcPPI10-siR2	619–639	GGGUGACGGUGCAUGCUAAUC	UUAGCAUGCACCGUCACCCCC	90.3
BcNmt1-siR1	599–619	CCCUCGUUCUGAUGUUCAAAU	UUGAACAUCAGAACGAGGGCG	100.2
BcNmt1-siR2	648–668	GGAUGUUGUUGCUUCUGUUCC	AACAGAAGCAACAACAUCCAA	102.3
BcAC-siR1	650–670	GGUAGAAAUUGAUGCUCAAGU	UUGAGCAUCAAUUUCUACCCU	104.4

## Data Availability

Data are contained within the article.

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
