# Peer review of "Development of an RNA Nanostructure for Effective Botrytis cinerea Control through Spray-Induced Gene Silencing without an Extra Nanocarrier"

_jof, 2024, doi:10.3390/jof10070483_

Round 1
Reviewer 1 Report
see above
Fangli Wu and co-authors report on the spray-induced gene silencing of a square shaped RNA against Botrytis cinerea. The data on this new RNA structure is timely and promising, however the description of the experimental design is lacking and the experiments fail to test for the additive effect of the 4 genes tested for the dsRNA. As currently shown the RNA square silencing 4 genes is compared to the dsRNA silencing of a single gene.
General comments:
Lots of abbreaviation that are not fully written such as PDA ( line 73), MS (line 114),
The strain that is used should be described. Is it B05.10, the isolate with the reference genome?
I suspect that a lot of misunderstandings in the methods come from an English language editor who did not understand the experiment. As currently written, the English is grammatically correct but doesn’t make sense experimentally.
Specific comments:
Line 62. The genes used here need to be described in more details, including Bcin gene ID
Line 73 + 131-133. Doesn’t make any sense. You grow Botrytis on PDA, described as the media for cultivation, but then write that spores for experiments came from infected tomato plants? How can you be sure that spores are only Botrytis, if not cultivated in vitro under sterile condition but collected from infected tomato plants? Do you mean that you collected the spores from the PDA culture, washed the spores, diluted to 100000 spores/ml and then inoculated on tomato?
Line 114: Did you grow Botrytis on Murashige & Skoog media for plants? Why?
Line 120: Was the plug from PDA culture?
Line 130: What does 7 day spraying regimen mean? Were the leaves sprayed everyday for a week? What was the time interval between spraying? What was the volume of RNA solution sprayed on each leaf each time?
Line 134-135: “This process was conducted over 3 days”. Given the previous sentence, that means you re-infected with Botrytis everyday for 3 days? That doesn’t make sense. Translation errors? If you really re-infected, why would you do that instead of letting Botrytis grow?
Line 137: What is the experimental design for this experiment? Was the bioassay replicated? How many different plants [biological replicates] were used?
Line 214-215: The design for the dsRNA is never described. What is the length of the dsRNA for DC1? Is it only 20 nucleotides or much longer? From Figure 2, it suggests that the dsRNA for DC1 is longer than the complete RNA square. Is it ~700bp long?
Line 235-247: The experimental design is incomplete
- The control of growth of the Botrytis strain without any RNA is missing. How can you assess the RNA reatment doesn’t have an effect on its own? The growth of the strain only on PDA should also be reported.
- dsRNA is only DC1, when the RNA square contain siRNA for 4 genes. To really assess the effect of the square, structure and effect of gene silencing, the dsRNA control should include the additive effect of all 4 genes as well.
Figure 2: what type of ladder is that? What are the maximum and minimum sizes? Is it a 1kb ladder?
Figure 3C, 4B & 4D: barplots are not an appropriate statistical representation of the data and should not be used.
Why: https://www.nature.com/articles/nmeth.2837
Please present a boxplot that represent the variance among the replicates.
Figure 4A: What is the experimental design? Were leaves sampled from different plants treated with RNA? Are the leaves from a single plant? Those details are essential to know if those are technical replicate or biological replicates.
Where the leaflets considered as leaves for tomato? Figure 4A present 5 leaflets infected. Does that mean only 2 tomato leaves were infected? Given the differences in leaf size, it is obvious from the pictures that some lesion could not develop fully because they reached the borders of the leaf.
Figure 4C/D: How do you explain that the dsRNA DC1 has larger lesion?
Line 272-273: Given that the comparison is between a square RNA silencing 4 genes compared to DC1 dsRNA only, the conclusion that square RNA is superior to dsRNA when the additive effect of the 4 genes was not tested is not appropriate.
Figure 5: How do you explain the relative expression of DC1 is minimal at 4dps and then increase with time?
Author Response
Reviewer 1’s comments:
Minor issues: - Lots of abbreaviation that are not fully written such as PDA ( line 73), MS (line 114). For media, the composition has to be clearly described. Therefore, did you grow Botrytis on Murashige & Skoog media for plants? - Line 62. The genes used here need to be described in more details, including Bcin gene ID. - The Botrytis strain that is used should be fully described. Is it B05.10, the isolate with the reference genome? The RNA treatment lack clarity: What does 7 day spraying regimen mean? Were the leaves sprayed everyday for a week? What was the time interval between spraying? What was the volume of RNA solution sprayed on each leaf each time? MAJOR issues: An important issue is the incoherence of the description of the growth and spore collection of Botrytis prior to infection: Doesn’t make any sense. You grew Botrytis on PDA, described as the media for cultivation, but then write that spores for experiments came from infected tomato plants? How can you be sure that spores are only Botrytis, if not cultivated in vitro under sterile condition but collected from infected tomato plants? Do you mean that you collected the spores from the PDA culture, washed the spores, diluted to 100000 spores/ml and then inoculated on tomato? Was the strain used isolated from tomato and then further cultivated on PDA? The design for the DC1 dsRNA is never described. What is the length of the dsRNA for DC1? Is it only 20 nucleotides or much longer? From Figure 2, it suggests that the dsRNA for DC1 is longer than the complete RNA square. Is it ~700bp long? -The experimental design is incomplete for the two experiments comparing NC-RNA, dsRNA and RNA square. The dsRNA is only DC1, when the RNA square contain siRNA for 4 genes. To really assess the effect of the square, structure and effect of all gene silenced, the dsRNA control should include the additive effect of all 4 genes as well.
Minor issues: - Lots of abbreaviation that are not fully written such as PDA ( line 73), MS (line 114).
Answer: Thank you for your comments. We conducted a thorough check of the entire manuscript and supplemented the full names of abbreviations that appeared for the first time in the text, including PDA and MS.
For media, the composition has to be clearly described. Therefore, did you grow Botrytis on Murashige & Skoog media for plants?
Answer: In the revised manuscript, we have added the components of MS medium (4.3 g/L MS basal salts supplemented with sucrose 1% (m/v), adjusted to pH 5.7 and solidified with 1.5% agar). In this study, we used MS medium to investigate the effect of RNA nanoparticles on the germination of Botrytis cinerea spores according to the report of Bilir et al. (Mol Plant Pathol. 2019).
- Line 62. The genes used here need to be described in more details, including Bcin gene ID. - The Botrytis strain that is used should be fully described. Is it B05.10, the isolate with the reference genome?
Answer: In the revised manuscript, we have added the details of these genes including names and gene IDs. In addition, we have also added the isolate information of Botrytis cinerea (B05.10) in the revised manuscript.
The RNA treatment lack clarity: What does 7 day spraying regimen mean? Were the leaves sprayed everyday for a week? What was the time interval between spraying? What was the volume of RNA solution sprayed on each leaf each time?
Answer: Thank you for your comments. Sorry, the description of RNA treatment in the section 2.7 is very unclear and confusing. In the revised manuscript, we rewrote this part as follows: Tobacco plants with uniform growth were selected for bioassay, and each of them was manually sprayed with 2 mL RNA solution at a concentration of 100 ng/µL. These RNA-sprayed plants were then grown in the greenhouse under a 16:8 h photoperiod at 24°C. Each RNAs were set up five parallel repetitions. At seven days post spraying (dps), a leaf per plant with uniform size was picked from each RNA-sprayed plant and then was inoculated with a 4 mm mycelia agar plug on the surface. The leaves without RNA treatment (No RNA) were inoculated with a mycelia agar plug as a negative control. Incubation was at 24°C under a 16:8 h photoperiod and 85% humidity. Subsequently, lesion sizes on the pathogen-infected plant materials were measured at two days post-inoculation (dpi).
MAJOR issues: An important issue is the incoherence of the description of the growth and spore collection of Botrytis prior to infection: Doesn’t make any sense. You grew Botrytis on PDA, described as the media for cultivation, but then write that spores for experiments came from infected tomato plants? How can you be sure that spores are only Botrytis, if not cultivated in vitro under sterile condition but collected from infected tomato plants? Do you mean that you collected the spores from the PDA culture, washed the spores, diluted to 100000 spores/ml and then inoculated on tomato? Was the strain used isolated from tomato and then further cultivated on PDA?
Answer: Thank you very much for your comments. Sorry, we made a mistake in the article. We have corrected this description (Please see the section 2.1 in revised manuscript). In addition, all confusing descriptions have been thoroughly corrected in the revised manuscript.
In the revised manuscript, we have improved the description of spore culture and collection as follows: Conidiospores were harvested from two-week-old PDA cultures, suspended in milli-Q water, and adjusted to a concentration of 5 × 106 spores/mL for subsequent analysis (Please see the Materials and Methods section 2.1). in addition, the inoculation of B. cinerea on RNA-sprayed leaves has used mycelia agar plug instead of spore solution (please see the Method section 2.7).
The design for the DC1 dsRNA is never described. What is the length of the dsRNA for DC1? Is it only 20 nucleotides or much longer? From Figure 2, it suggests that the dsRNA for DC1 is longer than the complete RNA square. Is it ~700bp long? -The experimental design is incomplete for the two experiments comparing NC-RNA, dsRNA and RNA square. The dsRNA is only DC1, when the RNA square contain siRNA for 4 genes. To really assess the effect of the square, structure and effect of all gene silenced, the dsRNA control should include the additive effect of all 4 genes as well.
Answer: Thank you for your comments. As your suggestion, we reconstructed and synthesized dsRNA targeting these four genes and added the construction details of this dsRNA in the revised manuscript as follows: a 100 bp gene-specific sequence was obtained from each of the four genes (DCL1, PPI10, NMT1 and AC) via blast searching against the mRNA library of B. cinerea. These four fragments were then connected into a 400 bp sequence, which was chemically synthesized and inserted into L4440 plasmid using Xba I and Hind III restriction enzyme sites. The sequence detail has been showed in in Table S1. (Please see section 2.3)
Major issue: - The use of barplot presenting only the mean of replicated samples is used. This is not acceptable for publication in 2024. Please plot the original data with a boxplot to show the variance across replicated. - The data reported is inconsistent: Fig 4B refers to necrotic area in % [of leaves?], when 4C report diameter in mm. Please report all data in lesion area in mm.
Answer: In the revised manuscript, we have replaced the original bar graph with a box plot. In addition, the lesion size in Figure 4b has also been updated by using lesion diameter (in mm) rather than lesion area.
Detail comments
Fangli Wu and co-authors report on the spray-induced gene silencing of a square shaped RNA against Botrytis cinerea. The data on this new RNA structure is timely and promising, however the description of the experimental design is lacking and the experiments fail to test for the additive effect of the 4 genes tested for the dsRNA. As currently shown the RNA square silencing 4 genes is compared to the dsRNA silencing of a single gene.
General comments:
Lots of abbreaviation that are not fully written such as PDA ( line 73), MS (line 114),
Answer: We conducted a thorough check of the entire manuscript and supplemented the full names of abbreviations that appeared for the first time in the text, including PDA and MS.
The strain that is used should be described. Is it B05.10, the isolate with the reference genome?
Answer: We have added the isolate information of Botrytis cinerea (B05.10) in the revised manuscript (Please see the section 2.1).
I suspect that a lot of misunderstandings in the methods come from an English language editor who did not understand the experiment. As currently written, the English is grammatically correct but doesn’t make sense experimentally.
Answer: We have carefully proofread the language of the entire article and corrected all sentences that may cause misunderstanding or make no sense.
Specific comments:
Line 62. The genes used here need to be described in more details, including Bcin gene ID
Answer: We have added the gene names and gene IDs in the revised manuscript.
Line 73 + 131-133. Doesn’t make any sense. You grow Botrytis on PDA, described as the media for cultivation, but then write that spores for experiments came from infected tomato plants? How can you be sure that spores are only Botrytis, if not cultivated in vitro under sterile condition but collected from infected tomato plants? Do you mean that you collected the spores from the PDA culture, washed the spores, diluted to 100000 spores/ml and then inoculated on tomato?
Answer: Thank you very much for your comments. Sorry, we made a mistake in the article. The Botrytis cinerea spores used in this study were from PDA plates. In the revised manuscript, we have corrected the description of line 73 and deleted the content of lines 131-133.
In the revised manuscript, we have improved the description of spore culture and collection as follows: Conidiospores were harvested from two-week-old PDA cultures, suspended in milli-Q water, and adjusted to a concentration of 5 × 106 conidiospores/mL for subsequent analysis (Please see the Materials and Methods section 2.1). In addition, the content about spore culture and collection has been deleted in Line 131-133.
Line 114: Did you grow Botrytis on Murashige & Skoog media for plants? Why?
Answer: For the spore germination assays in section 2.5, we followed the method reported by Bilir et al. (Mol Plant Pathol. 2019) and indeed used MS as the culture medium. Actually, we found that the spores of Botrytis cinerea could germinate normally even in water, so any culture medium can be used for the spore germination assays.
Line 120: Was the plug from PDA culture?
Answer: Yes, agar plugs containing active mycelium of B. cinerea were collected from a 7-day-old PDA culture. We have improved the plug information in the revised manuscript.
Line 130: What does 7 day spraying regimen mean? Were the leaves sprayed everyday for a week? What was the time interval between spraying? What was the volume of RNA solution sprayed on each leaf each time? Line 134-135: “This process was conducted over 3 days”. Given the previous sentence, that means you re-infected with Botrytis everyday for 3 days? That doesn’t make sense. Translation errors? If you really re-infected, why would you do that instead of letting Botrytis grow? Line 137: What is the experimental design for this experiment? Was the bioassay replicated? How many different plants [biological replicates] were used?
Answer: Thank you very much for your comments. We have rewritten the section 2.7 in revised manuscript containing lines 133-137. Our updates are as follows: Tobacco plants with uniform growth were selected for bioassay, and each of them was manually sprayed with 2 mL RNA solution at a concentration of 100 ng/µL. These RNA-sprayed plants were then grown in the greenhouse under a 16:8 h photoperiod at 24°C. Each RNAs were set up five parallel repetitions. At seven days post spraying (dps), a leaf per plant with uniform size was picked from each RNA-sprayed plant and then was inoculated with a 4 mm mycelia agar plug on the surface. The leaves without RNA treatment (No RNA) were inoculated with a mycelia agar plug as a negative control. Incubation was at 24°C under a 16:8 h photoperiod and 85% humidity. Subsequently, lesion sizes on the pathogen-infected plant materials were measured at two days post-inoculation (dpi)
Line 214-215: The design for the dsRNA is never described. What is the length of the dsRNA for DC1? Is it only 20 nucleotides or much longer? From Figure 2, it suggests that the dsRNA for DC1 is longer than the complete RNA square. Is it ~700bp long?
Answer: Thank you for your comments. In the revised manuscript, we reconstructed and synthesized dsRNA targeting four genes and added the construction details of this dsRNA in the section 2.3 as follows: a 100 bp gene-specific sequence was obtained from each of the four genes (DCL1, PPI10, NMT1 and AC) via blast searching against the mRNA library of B. cinerea. These four fragments were then connected into a 400 bp sequence, which was chemically synthesized and inserted into L4440 plasmid using Xba I and Hind III restriction enzyme sites. The sequence detail has been showed in in Table S1.
Line 235-247: The experimental design is incomplete- The control of growth of the Botrytis strain without any RNA is missing. How can you assess the RNA reatment doesn’t have an effect on its own? The growth of the strain only on PDA should also be reported. - dsRNA is only DC1, when the RNA square contain siRNA for 4 genes. To really assess the effect of the square, structure and effect of gene silencing, the dsRNA control should include the additive effect of all 4 genes as well.
Answer: Thank you for your comments. As your suggestion, in the revised manuscript, we reconstructed and synthesized a dsRNA targeting these four genes. We then redid the experiments using the newly obtained dsRNA and updated all the experimental results. At the same time, we also added a control without RNA treatment to evaluate the germination, growth and infection of Botrytis cinerea in the absence of RNA treatment.
Figure 2: what type of ladder is that? What are the maximum and minimum sizes? Is it a 1kb ladder?
Answer: In the revised manuscript, we have marked that the DNA marker used in Figure 2 is DL2000, and noted the sizes of some bands.
Figure 3C, 4B & 4D: barplots are not an appropriate statistical representation of the data and should not be used. Why: https://www.nature.com/articles/nmeth.2837
Please present a boxplot that represent the variance among the replicates.
Answer: In the revised manuscript, we have replaced the original bar graph with a box plot.
Figure 4A: What is the experimental design? Were leaves sampled from different plants treated with RNA? Are the leaves from a single plant? Those details are essential to know if those are technical replicate or biological replicates.
Answer: Thank you for your comments. In the revised manuscript, we have rewritten the content in Methods section 2.7, which contains the experimental design and process of Figure 4. All above issues have been clearly explained in the revised section 2.7.
Where the leaflets considered as leaves for tomato? Figure 4A present 5 leaflets infected. Does that mean only 2 tomato leaves were infected? Given the differences in leaf size, it is obvious from the pictures that some lesion could not develop fully because they reached the borders of the leaf.
Answer: Thank you for your comment. Due to the small size of tomato leaves, lesions can easily reach the borders of the leaf. It is not easy to get clear differences in lesion size between different RNA treatments. Therefore, in the revised manuscript, we abandoned tomato leaves and only used larger tobacco leaves for bioassay analysis.
Figure 4C/D: How do you explain that the dsRNA DC1 has larger lesion?
Answer: Thank you for your comments. In the revised manuscript, we reconstructed dsRNA targeting four genes and found that the diameter of lesions on the leaves treated with dsRNA was smaller than that on the leaves treated with Nc-RNA or without RNA treatment, but larger than that on the leaves treated with RNA nanoparticles. So this problem no longer exists.
Line 272-273: Given that the comparison is between a square RNA silencing 4 genes compared to DC1 dsRNA only, the conclusion that square RNA is superior to dsRNA when the additive effect of the 4 genes was not tested is not appropriate.
Answer: Thank you for your comments. As your suggestion, we reconstructed and synthesized a dsRNA targeting these four genes in the revised manuscript, and then redid the experiments using the newly obtained dsRNA and updated all the experimental results. The latest results are consistent with previous results, showing that RNA squares have a stronger and longer-lasting inhibitory effect on Botrytis cinerea than dsRNA.
Figure 5: How do you explain the relative expression of DC1 is minimal at 4dps and then increase with time?
Answer: In the revised manuscript, we have added stem-loop RT-qPCR to detect the level of a DCL1-targeted siRNA. The results confirmed that the level of the siRNA was highest at 4 dps, so the level of DCL1 was lowest at that time.
Reviewer 2 Report
Dear Authors,
You have presented a manuscript entitled "Development of an RNA nanostructure for effective Botrytis cinerea control through spray-induced gene silencing without an extra nanocarrier", which explores the application of RNA nanoparticles for controlling the pathogen Botrytis cinerea in crops using the spray-induced gene silencing (SIGS) technology. Your study proposes a novel approach by designing RNA nanoparticles that are more stable and effective without requiring additional nanocarriers, potentially offering safer and more efficient alternatives for agricultural applications.
The introduction of your manuscript adequately sets the stage for the necessity of developing safer and more effective crop protection strategies, focusing on SIGS as an alternative to traditional chemical-based methods. The literature cited provides a sound basis for the challenges associated with RNA instability and the low efficacy of RNAi induction when using direct dsRNA applications. However, a more detailed discussion on the specific molecular mechanisms of SIGS and how RNA nanoparticles overcome these mentioned challenges could further enrich this section.
The methods section is detailed appropriately, allowing for the reproducibility of the study. However, the inclusion of more rigorous controls to compare the efficacy of RNA nanoparticles against existing treatments beyond dsRNA, such as chemical controls, would enhance the robustness of the findings.
Results demonstrate a superior inhibition of B. cinerea growth and spore germination by RNA nanoparticles compared to dsRNA, along with enhanced protection of treated plants. These results are promising and well presented with significant statistical data. Extending these studies to more plant types and environmental conditions to validate the robustness of RNA nanoparticles would be beneficial.
The discussion intelligently contextualizes the relevance of RNA nanoparticles in broader fields of nanotechnology and biotechnology, highlighting their stability and efficacy. A more detailed comparison with other emerging technologies and a discussion of potential regulatory or public perception concerns could provide a more comprehensive view of the potential and limitations of RNA nanoparticles.
While the conclusions effectively summarize the study and reaffirm the potential of RNA nanoparticles to improve pathogen control in agriculture sustainably, outlining future research directions, including long-term studies and environmental impact assessments, would be advisable.
In my critical opinion, manuscript is well-constructed and presents significant advancements in the field of agricultural biotechnology. However, the variability in the response of tomato plants as indicated by the large standard deviations in some experimental results calls for a more detailed statistical analysis to confirm the significance of the findings.
Revisions needed:
- Enhanced Statistical Analysis: Include a more detailed statistical analysis addressing the observed variability. Performing post-hoc tests following ANOVA to individually compare all conditions could provide clearer insights into the differences between treatments.
- Increased Replication: Increasing the number of experimental replicates could help stabilize variance and provide more accurate effect estimates of the treatments.
- Uniformity in Treatment Application: Ensure that the treatment application across all samples is uniform to minimize variability due to application differences.
Considering the innovative approach and the potential impact of your findings, I recommend major revision before the manuscript can be accepted for publication.
Sincerely,
No minor revision are needed at the moment.
Author Response
Dear Authors,
You have presented a manuscript entitled "Development of an RNA nanostructure for effective Botrytis cinerea control through spray-induced gene silencing without an extra nanocarrier", which explores the application of RNA nanoparticles for controlling the pathogen Botrytis cinerea in crops using the spray-induced gene silencing (SIGS) technology. Your study proposes a novel approach by designing RNA nanoparticles that are more stable and effective without requiring additional nanocarriers, potentially offering safer and more efficient alternatives for agricultural applications.
The introduction of your manuscript adequately sets the stage for the necessity of developing safer and more effective crop protection strategies, focusing on SIGS as an alternative to traditional chemical-based methods. The literature cited provides a sound basis for the challenges associated with RNA instability and the low efficacy of RNAi induction when using direct dsRNA applications. However, a more detailed discussion on the specific molecular mechanisms of SIGS and how RNA nanoparticles overcome these mentioned challenges could further enrich this section.
Answer: In the revised manuscript, we have added a discussion on the molecular mechanism of SIGS and discussed why the silencing efficiency of the target gene is highest on the fourth day rather than on the first day after RNA nano spraying. The possible reason is due to RNA nanoparticles have stronger stability and are more tolerant to BcDCL, the RNA square taken up by the B. cinerea were not completely processed into siRNA within one day. Within four days after spraying, they were gradually consumed, and the siRNA level was increased, resulting in the lowest expression level of the target gene appearing on the fourth day after spraying. However, starting from the 4th day after spraying, no new siRNA will be generated in the B. cinerea because the RNA nanoparticles have been completely consumed, resulting in a decrease in the level of RNA square-derived siRNAs in the B. cinerea over time. Therefore, starting from the 4th day after the RNA nanoparticles were sprayed, the expression level of the target gene in the gray mold fungus gradually increased.
The methods section is detailed appropriately, allowing for the reproducibility of the study. However, the inclusion of more rigorous controls to compare the efficacy of RNA nanoparticles against existing treatments beyond dsRNA, such as chemical controls, would enhance the robustness of the findings.
Answer: Thank your for your comments. We carefully checked the research methods involved in this study. We supplemented and improved the research methods that were not clearly described to ensure the reproducibility of the study.
In addition, we also tried to add carbendazim as another control. According to the reports by Cong et al (Plant Dis. 2018), the compound carbendazim is often used to control B. cinerea, and the concentration is generally 50-200 ng/ul. However, in our experiment, it was found that the growth of B. cinerea mycelium plug inoculated on the PDA plate coated with carbendazim at a concentration of 500 ng/ul was similar to that on the PDA plate without RNA treatment, suggesting that the carbendazim we used had no inhibitory effect on the growth of gray mold. We know this is definitely wrong, but we don’t know where the problem lies.
The antifungal experiment of chemical agents is usually conducted on experimental filter paper discs, and the antifungal effect is determined by analyzing the size of the inhibition zone. In this study, we used RNA for antifungal testing by directly coating the RNA on the surface of the PDA plate, which is different from the antifungal zone method. Perhaps this difference is a possible reason for the different results. But in any case, it will take a lot of time to test and establish appropriate methods to use chemical agents as a control in SIGS analysis. It is difficult for us to complete the establishment of these methods within the limited paper revision period.
Given that the focus of this study is to develop a more stable RNA form for the control of Botrytis cinerea, it provides a new idea for plant protection strategies based on SIGS. Therefore, it is definitely better to use chemical agents as a control in the study, but it is also acceptable not to use them. Based on the above reasons, we did not add chemical agents as a control in the revised manuscript.
Results demonstrate a superior inhibition of B. cinerea growth and spore germination by RNA nanoparticles compared to dsRNA, along with enhanced protection of treated plants. These results are promising and well presented with significant statistical data. Extending these studies to more plant types and environmental conditions to validate the robustness of RNA nanoparticles would be beneficial.
The discussion intelligently contextualizes the relevance of RNA nanoparticles in broader fields of nanotechnology and biotechnology, highlighting their stability and efficacy. A more detailed comparison with other emerging technologies and a discussion of potential regulatory or public perception concerns could provide a more comprehensive view of the potential and limitations of RNA nanoparticles.
Answer: In the discussion section of the revised manuscript, we added a mechanistic discussion of SIGS and pointed out that, like other nanocarriers, RNA nanoparticles can effectively improve the efficiency of cellular uptake of exogenous siRNA and the persistence of induced RNAi (Please see the second paragraph of the discussion section). In addition, unlike conventional nanomaterials that are only used as delivery carriers, the RNA nanoparticles used in this study are also triggers for RNAi. After entering the organism, they will be completely consumed and will not accumulate in the organism. Therefore, compared with other nanomaterials, RNA nanoparticles have less environmental risks and higher safety.
While the conclusions effectively summarize the study and reaffirm the potential of RNA nanoparticles to improve pathogen control in agriculture sustainably, outlining future research directions, including long-term studies and environmental impact assessments, would be advisable.
Answer: Thank you very much for your comments. According to your suggestion, we have added the future research directions of RNA nanoparticles and their potential applications in SIGS-based plant protection in the conclusion.
In my critical opinion, manuscript is well-constructed and presents significant advancements in the field of agricultural biotechnology. However, the variability in the response of tomato plants as indicated by the large standard deviations in some experimental results calls for a more detailed statistical analysis to confirm the significance of the findings.
Revisions needed:
- Enhanced Statistical Analysis: Include a more detailed statistical analysis addressing the observed variability. Performing post-hoc tests following ANOVA to individually compare all conditions could provide clearer insights into the differences between treatments.
- Increased Replication: Increasing the number of experimental replicates could help stabilize variance and provide more accurate effect estimates of the treatments.
- Uniformity in Treatment Application: Ensure that the treatment application across all samples is uniform to minimize variability due to application differences.
- Considering the innovative approach and the potential impact of your findings, I recommend major revision before the manuscript can be accepted for publication.
Answer: In the revised manuscript, we re-analyzed all the data using one-way ANOVA followed by Tukey’s test. Details of the statistical analysis are described in Methods 2.10.
Since we redesigned and obtained new dsRNA according to the suggestions of reviewer 1, all experiments need to be redone, which is equivalent to adding one more repetition. In the experimental results obtained using the new dsRNA, RNA squares still have a stronger and more lasting inhibitory effect on gray mold than dsRNA, which is completely consistent with the results of the previous version.
During the manuscript revision process, we paid great attention to the consistency of experimental treatments to ensure the reliability and repeatability of the research results.
Finally, the innovation of this study and its potential application in future SIGS-based control were added in the discussion and conclusion.
Sincerely,
Round 2
Reviewer 1 Report
The authors fixed most of the experimental descriptions that were missing or not accurate.
However, the dsRNA treatment is still not clearly described. The authors say that now they did the dsRNA for all target genes, but they do not present the results. Also when asked about the tomato results, they remove those results rather than explaining what is going on.
What is the dsRNA treatment in figure 3 and 4? Is it the mix of all 4 genes for the dsRNA? It remains unknown,
My main concern is that the results changed between the first submission and revisions.
The 1st: dsRNA in fig 3 was at ~70mm and RNA square ~50mm. In the revision figure 3: dsRNA is ~50mm and RNA square ~20mm. It would make sense if the 1st submission dsRNA was only dc1 and in the second the dsRNA is now the mix of all 4 dsRNAs. Therefore the silencing effect is larger. However, why did the RNA square decrease from 50 to 20mm? This seems highly convenient as otherwise the conclusion of the whole manuscript that the RNA square is more effective is void.
Same for the germination, RNA square went from some germination to no germination in image 3a.
See comments to authors
Reviewer 2 Report
Dear Authors,
Thank you for submitting your revised manuscript titled "Development of an RNA Nanostructure for Effective Botrytis cinerea Control through Spray-Induced Gene Silencing without an Extra Nanocarrier."
After reviewing the improvements made based on my previous comments, I am pleased to inform you that I consider the paper to be acceptable for publication.
Thank you for addressing the suggestions and enhancing the quality of your work. I appreciate your efforts and look forward to seeing your article published.
Best regards,
No minor details are needed.
Author Response
Thank you very much for your comment.